# Partial Versus Total Omentectomy in Patients with Gastric Cancer: A Systemic Review and Meta-Analysis

**DOI:** 10.3390/cancers13194971

**Published:** 2021-10-03

**Authors:** Shion Wei Chai, Suo-Hsien Wang, Chih-Yuan Wang, Yi-Chan Chen, Ruey-Shyang Soong, Ting-Shuo Huang

**Affiliations:** 1Division of General Surgery, Department of Surgery, Chang Gung Memorial Hospital, Keelung Branch, No. 222, Mai-Chin Road, Keelung 20401, Taiwan; swchai1988@cgmh.org.tw (S.W.C.); b9802008@cgmh.org.tw (S.-H.W.); drwcyuan@cgmh.org.tw (C.-Y.W.); 8902073@cgmh.org.tw (Y.-C.C.); raymond@cgmh.org.tw (R.-S.S.); 2Department of Chinese Medicine, College of Medicine, Chang Gung University, Kwei-Shan, Taoyuan 259, Taiwan; 3Community Medicine Research Center, Chang Gung Memorial Hospital, Keelung 20401, Taiwan

**Keywords:** gastric cancer, gastrectomy, total omentectomy, partial omentectomy, survival

## Abstract

**Simple Summary:**

Gastric cancer is one of the leading causes of cancer-related mortality, especially in Asia. Radical gastrectomy, including omentectomy, is the standard surgical procedure for curative treatment. Nevertheless, total omentectomy may have an impact on postoperative complications. Although the omentum serves as a bridge for peritoneal metastasis, some clinicians propose that the omentum could participate in anti-bacterial defense, hemostasis, and prevention of intestinal adhesions. Clinically, it is controversial whether extensive omentectomy provides better survival to patients. Here, we conducted a systematic review and meta-analysis to investigate the safety and efficacy of partial omentectomy compared to total omentectomy during radical gastrectomy in patients with gastric cancer. We demonstrate that partial omentectomy has non-inferior long-term oncological outcomes compared to total omentectomy. In addition, partial omentectomy is associated with shorter operative time and lesser blood loss. Therefore, it may not be necessary to perform total omentectomy routinely.

**Abstract:**

*Background:* Surgical treatment is the key to cure localized gastric cancer. There is no strong evidence that supports the value of omentectomy. Thus, a meta-analysis was conducted to compare the safety and efficiency of partial and total omentectomy in patients with gastric cancer. *Methods:* PubMed, Embase, and Cochrane Library databases were searched. All studies that compared total and partial omentectomy as treatments for gastric cancer were included. The primary outcomes were patients’ overall survival and disease-free survival, while the secondary outcomes were perioperative outcome and postoperative complications. *Results:* A total of nine studies were examined, wherein 1043 patients were included in the partial omentectomy group, and 1995 in the total omentectomy group. The partial omentectomy group was associated with better overall survival (hazard ratio: 0.80, 95% CI: 0.66 to 0.98, *p* = 0.04, *I^2^* = 0%), shorter operative time, and lesser blood loss than the total omentectomy group. In addition, no statistically significant difference was observed in the number of dissected lymph nodes, length of hospital stays, complication rate, and disease-free survival. *Conclusions:* Our results show that, compared with total omentectomy in gastric cancer surgery, partial omentectomy had non-inferior oncological outcomes and comparable safety outcomes.

## 1. Introduction

Gastric cancer (GC) is the fourth most common cancer worldwide. In 2020, a total of 1,089,103 (5.6% of all cancer) new gastric cancer cases, causing 768,793 deaths (7.7% of all cancer), were estimated [1]. Although its global incidence has been declining, GC is still one of the leading causes of cancer-related mortality, especially in Asia. 

Surgical treatment is the key to cure localized GC. In various international guidelines, D2 lymph node dissection is generally recommended during gastrectomy. Theoretically, both omentum and bursa omentalis should be resected to prevent peritoneal metastasis. However, a recent meta-analysis has shown that gastrectomy with bursectomy is not superior in terms of survival to gastrectomy without bursectomy and is thus bursectomy not recommend as a standard treatment modality for cT3 and cT4 GC [2]. Likewise, although the omental lymph system was suggested as a bridge for metastasis to the peritoneal cavity in animal models [3], and omental lavage occasionally detect omental micrometastasis in patients with GC [4], there is still no evidence showing a definitive improvement of survival after gastrectomy with total omentectomy.

Various clinical practice guidelines have acknowledged the issue of whether gastrectomy with omentectomy should be performed. According to the Japanese GC treatment guidelines, omentectomy is recommended in standard gastrectomy for T3 or deeper GC [5]. Similarly, the National Comprehensive Cancer Network (NCCN) Guidelines suggest resection of both the greater and the lesser omentum during D1 dissection. However, the European Society for Medical Oncology guidelines do not mention omentectomy in the treatment of GC [6]. 

The greater omentum is a free-hanging mesenteric tissue curtain in the abdominal cavity. Recent advances show its angiogenic functions and the presence of lymphatics; the omentum has been used in reconstructive surgeries, including for defect coverage, vascular pedicles, and vascularized lymph node transplant. In addition, the omentum may perform many physiological functions, being involved in antibacterial defense, hemostasis, and prevention of intestinal adhesions [7,8,9].

In the past, it was thought of as a useless, large, mesenterial fold and was often sacrificed by surgeons during abdominal surgery. However, such extensive dissection of the omentum or bursa omentalis may carry risks of injuries to the adjacent organs such as the colon, spleen, and pancreas. Furthermore, extensive omentectomy during radical gastrectomy may be complicated by bleeding, anastomosis dehiscence, and infections. Thus, the extent of omentectomy may have a significant impact on intraoperative and postoperative complications.

On the other hand, laparoscopic radical gastrectomy is now a trending method in Asia and western countries. However, total omentectomy is a time-consuming procedure and may increase the risk of complications. Therefore, we conducted a systematic review and meta-analysis to investigate the safety and efficacy of partial omentectomy compared with total omentectomy in terms of survival outcomes, perioperative outcomes, and postoperative complications in patients with GC.

## 2. Materials and Methods

PROSPERO registration number: CRD42021273755.

### 2.1. Literature Search Strategy

Relevant studies were found using computerized searches in the PubMed, EMBASE, and Cochrane Library databases. The following terms were used for MeSH and free-text searching: gastric cancer, omentectomy, and oment*. All relevant articles and reference lists in the selected studies were obtained from the above databases. Articles were searched regardless of their language. The final search was performed in April 2021.

### 2.2. Data Extraction

The inclusion of trials in this study was decided upon independently by two reviewers (SWC, SHW). The studies’ data regarding study design, participant characteristics, inclusion and exclusion criteria, surgical techniques, operation time, blood loss, number of harvested lymph nodes (LN), complications, hospital stay, disease-free survival (DFS), and overall survival (OS) were extracted. Inconsistencies between the data collected by the two reviewers were resolved by a third reviewer (CYW).

### 2.3. Inclusion and Exclusion Criteria

Prospective and retrospective comparative studies and randomized controlled trials (RCTs) that evaluated the outcomes of partial omentectomy and total omentectomy in patients with GC were included in this study. The inclusion criteria were as follows: (1) studies involving patients with histologically proven primary adenocarcinoma of the stomach and (2) studies involving patients with no other malignancy in the past five years. Non-comparative review, case series, and case reports were excluded. 

### 2.4. Assessment of The Quality of The Methods

RCTs were assessed using the Cochrane Collaboration’s risk of bias tool for RCTs [10], while retrospective studies were assessed using the Newcastle–Ottawa Scale (NOS) [11]. We defined “high quality” as an NOS score of ≥8 points and “moderate quality” as an NOS score between 5 and 7 points. Two reviewers independently assessed the eligibility of each trial. Disagreement between the two was resolved through discussion and consensus. 

### 2.5. Statistical Analyses

The primary outcomes were OS and DFS. The secondary outcomes included operation time, blood loss, numbers of harvested LNs, complications, and length of hospital stay. As for complications, we extracted complication data using the Clavien–Dindo Classification. Numbers of Clavien–Dindo Classification ≥3 (composite outcomes) between the 2 groups in primary studies were retrieved for meta-analysis. The meta-analysis was performed according to the Preferred Reporting Items for Systematic Reviews (PRISMA) guidelines [12]. Hazard ratios (HRs) with corresponding 95% confidence intervals (CIs) were used as the summary measure to compare OS and DFS between the 2 groups in the meta-analysis. When HRs with 95% CIs were not reported in primary studies, we extracted this information from Kaplan–Meier curves presented in the included studies, using Tierney et al.’s previously published methods [13]. Treatment of dichotomous outcomes was summarized as odd ratios (OR). Since in the included studies, several continuous outcomes were reported as medians with a range, we converted the data into mean and standard deviation using Hozo et al.’s previously published equation [14]. Continuous outcomes were reported as mean differences, with 95% CIs. The included population had different surgical approaches, different stages of GC, and other sources of variance; therefore, the random effects model was used. We assessed clinical heterogeneity by comparing the methodologies of the included studies. Statistical heterogeneity of effect sizes between studies was assessed using the *I^2^* statistic and the Cochrane Q test (known as the chi-squared test). The possibility of publication bias was evaluated by plotting effect size vs. corresponding standard errors. We defined statistical heterogeneity using a cut-off value of *p* ≤ 0.10 for the Cochrane Q test results or *I^2^* ≥ 50%. All analyses were performed using R (version 3.6.3) (R Foundation for Statistical Computing, Vienna, Austria) with the contributed packages “meta” [15], “metafor” [16], and “lattice” [17]. R coding was provided in Appendix A. 

## 3. Results

### 3.1. Included Trials

Figure 1 shows the processes for screening and selecting the trials in the PRISMA flow diagram. We identified 513 records through the database search. After screening titles and abstracts, 468 non-relevant studies were removed. In addition, duplicated studies, comments, non-comparative reports, different comparative studies, and ongoing clinical trials were excluded. A total of nine studies [18,19,20,21,22,23,24,25,26] that examined partial omentectomy and total omentectomy during radical gastrectomy were included in the final analysis. Among these studies, eight retrospective studies, including four propensity score-matching studies and one randomized controlled study were included in this meta-analysis. We summarize the study characteristics and patient demographic data for each study in Table 1. Table 2 shows the results of the quality assessment of the studies included in this meta-analysis. Four of eight non-randomized comparative studies were rated as “high-quality”, whereas other four studies were classified as having “moderate quality”. All pooled results are summarized in Table 3.

### 3.2. Effects of the intervention

#### 3.2.1. Primary outcomes

Five studies reported data about DFS. The results of the meta-analysis revealed no statistically significant difference between the two groups (HR: 0.84, 95% CI: 0.69 to 1.01, *p* = 0.06, I^2^ = 0%) (Figure 2). Subgroup analysis according to the study design found that four PSM studies had similar results (HR: 0.85, 95% CI: 0.71 to 1.04, *p* =0.11, I^2^ = 0%). Kim MC et al. [20] found no recurrence between the two groups in the follow-up of 38.1 months among patients with early gastric cancer. For OS, Kim DJ et al. [22] reported disease-specific OS; however, the other five studies reported OS. We pooled these six studies and found there was statistically significant differences between the two groups. (HR: 0.80, 95% CI: 0.66 to 0.99, *p* = 0.04, I^2^ = 0%) (Figure 3). Subgroup analysis of four PSM studies provided similar result (HR: 0.79, 95% CI: 0.64 to 0.98, I^2^ = 0%). The funnel plots were symmetric for OS (Appendix A). For DFS, some small studies with negative effects might be missing. Thus, the true effect might be smaller than the observed effect if these studies exist (Appendix A).

#### 3.2.2. Secondary Outcomes

Six studies reported the operative time. Kim DJ et al [22]. reported the time needed for omentectomy. Analysis of the pooled data of seven studies revealed that the operative time was significantly shorter for the partial omentectomy group (MD: –16.72 mins, 95% CI: –28.80 to –4.6, *p* < 0.01, I^2^ = 95%). Subgroup analysis according to the operation method was done. Four studies using the open method revealed that the operative time was not significantly different (MD: –2.1mins, 95% CI: –10.9 to 6.7, *p* =0.64, I^2^ = 68%). Three studies reporting data related to the laparoscopic or the mixed method revealed that the operative time was significantly shorter for the partial omentectomy group (MD: –32.8 mins, 95% CI: –48.9 to –16.8, *p* < 0.01, I^2^ = 94%). Four studies mentioned the estimated blood loss. The summarized results indicated that intraoperative blood loss was significantly less for the partial omentectomy group (MD –95.3 mL, 95% CI –139.81 to –50.8, *p* < 0.01, I^2^ = 96%) than for the total omentectomy group. However, the differences in the number of LNs harvested and the length of stay were not statistically significant between the two groups. 

Kim DJ et al. [22] reported two intraoperative complications (splenic and mesocolon injury, requiring concurrent splenectomy and transverse colectomy) related to omentectomy in the total omentectomy group. Six studies mentioned adhesion- and ileus-related complications. Analysis of the pooled data revealed no significant difference between the two groups. (OR: 0.58, 95% CI: 0.31 to 1.09, *p* = 0.09, I^2^ = 0%) (Figure 4) Five studies reported postoperative complications. A meta-analysis of the complications revealed no statistically significant difference between the two groups. (OR: 0.85, 95% CI: 0.60 to 1.21, *p* = 0.37, I^2^ = 50%) 

#### 3.2.3. Additional Analyses

We conducted additional analyses to investigate the effects of the surgical approaches on OS and DFS. Forest plots demonstrated that OS and DFS were not significantly different between the partial omentectomy and the total omentectomy groups (Appendix A and Appendix A). Stratified analyses showed no significant heterogeneity for OS and DFS. 

## 4. Discussion

In the present study, we aimed to analyze the efficacy and safety of partial omentectomy compared with total omentectomy during radical gastrectomy in patients with GC. Our included studies defined partial omentectomy as the procedure dividing the greater omentum about 3–5 centimeters away from the gastroepiploic arcade and preserving the greater omentum on the transverse colon side. In total omentectomy, the gastrocolic ligament was detached from the transverse colon along the avascular plane. In our meta-analysis, the OS of GC patients who underwent partial omentectomy compared with that of patients subjected to total omentectomy during radical gastrectomy was significantly better. This finding might be related to the fact that patients with more advanced stages of GC were more likely to receive total omentectomy. DFS was not statistically significant between the two groups. All six included studies involved patients with locally advanced GC. Although these six studies were retrospective, four of them used the PSM method to minimize confounding biases. Complications in the two groups were similar. 

Recently, Ishizuka M et al. [27] published a meta-analysis of long-term postoperative outcomes in patients with locally advanced GC. They reported no significant difference in 5-year recurrence-free survival (RR, 0.91; 95% CI: 0.74–1.13; *p* = 0.41; *I^2^* = 0%) in the pooled data from PSM studies and in 5-year OS (RR, 0.77; 95% CI: 0.52–1.13; *p* = 0.18; *I^2^* = 47%) in the pooled data from two PSM studies between patients not undergoing and undergoing omentectomy. The studies included were different from the ones we examined. These authors included three studies [28,29,30] that were published in Japanese. These three studies included patients recruited between 1981 and 2001. In addition, patients with omentobursectomy were present in the control groups. Further, the total omentectomy group had a higher rate of total gastrectomy and a higher rate of extensive LN dissection than the partial omentectomy group. On the contrary, we included studies comparing GC patients subjected to partial omentectomy with those subjected to total omentectomy without bursectomy. Furthermore, we used the HR as the pooled effect, which is the preferred summary measure for studies reporting time-to-event outcomes with different follow-up times. However, the reliability of results using HR as the effect size relies on the validity of proportional hazards assumptions in primary studies. Our meta-analysis showed shorter operative time and lesser blood loss in the partial omentectomy group than in the total omentectomy group during GC treatment, which was similar with the results of the study published by the Ishizuka M et al. However, the two studies show substantial statistical heterogeneity in operative time, blood loss, total number of lymph nodes harvested, and length of stay. Such heterogeneity might be due to several reasons. First, most of the examined studies regarded patients with different stages of GC. Second, surgical procedures were quite different in these studies, as regards surgical approaches (open or laparoscopic) and anastomosis methods. Third, the improved and wide use of energy devices in recent years might be another source of heterogeneity. Fourth, operative time and blood loss might be reduced in high-volume centers. Finally, measurement errors were not uncommon in retrospective studies. 

We also hypothesize that partial omentectomy may have significant advantages compared to total omentectomy in laparoscopic settings with respect to open surgery in terms of operation time and surgical complications. We included five studies with the open approach, three studies with the mixed approach, and one study with the laparoscopic approach. Stratified analyses by surgical approaches revealed that OS and DFS were not significantly different between the partial omentectomy and the total omentectomy groups. Operation time was shorter for partial omentectomy compared to total omentectomy in the laparoscopic setting. On the other hand, there was no significant difference in the open settings. Nevertheless, we had limited data regarding blood loss, number of LNs harvested, ileus complications, and length of hospital stay in the pure laparoscopic study.

Lymphatic dissemination in GC is complex. Peritoneal carcinomatosis was common in patients with advanced GC undergoing radical gastrectomy. It is believed that total omentectomy and even bursectomy may improve patients’ survival; however, our analysis revealed similar DFS and rather better OS in the partial omentectomy group. In addition, the number of LNs harvested was comparable in both groups. Our included studies analyzed the first recurrent organ and found that the frequency of peritoneal carcinomatosis was similar in both groups. Long-term oncological results of the partial omentectomy group were not inferior to those of the total omentectomy group. With the improvement in adjuvant chemotherapeutic regimens, target therapy, and immunotherapy, patients with advanced-stage GC could be managed better either preoperatively or postoperatively. Thus, peri-operative adjuvant therapies such as treatment duration and compliance with adjuvant chemotherapy might be more important than the extent of omentectomy [31].

Only one of our included studies, Hasegawa et al.’s, mentioned neoadjuvant chemotherapy. Less than 5% of their cohorts received neoadjuvant therapy. For studies that did not include specific neoadjuvant chemotherapy, patients with a good response might have a good prognosis regardless of whether they received total or partial omentectomy. However, for patients with a poor repose to neoadjuvant chemotherapy, extensive dissection may be needed. Five of the included studies did not provide information on neo/adjuvant chemotherapy, and two studies enrolled patients with early gastric cancer. On the other hand, four studies described patients who received adjuvant chemotherapy, i.e., 340/659 (51.6%) in the partial omentectomy group and 416/659 (63.1%) in the total omentectomy group. Therefore, detailed information on preoperative and postoperative cancer treatment is needed to draw a solid conclusion. 

Radical gastrectomy may have complications, including bleeding, leakage, and infection. Patients with postoperative morbidities may have a worse prognosis. Indeed, postoperative complications might affect patients’ survival either directly or indirectly, such as delayed adjuvant chemotherapy. In the present study, five studies reported composite complications (Clavien–Dindo classification ≥3); the pooled result revealed no significant difference between the partial omentectomy group and the total omentectomy group (OR: 0.85, CI: 0.60 to 1.21, *p* = 0.37). Interestingly, many studies investigated risk factors for anastomotic leakage and how to avoid it. These studies demonstrated that Charlson co–morbidity index, Eastern Cooperative Oncology Group performance score and American Society of Anesthesiologists score, serum albumin levels, and psoas muscle volume might be associated with increased risks for anastomotic leakage. Recently, Radulescu et al. showed increased preoperative value for the neutrophil/lymphocyte ratio as an excellent negative predictor of anastomotic leakage [32]. Patients with at risk for anastomotic leakage might benefit from partial omentectomy because residual omentum might prevent adhesion of the intestines and migrate to cover the anastomosis sites. Although our analysis showed that the differences in adhesion and ileus-related complications were not statistically significant between the two groups (OR: 0.58, CI: 0.31 to 1.09, *p* = 0.09), the present study may not have adequate statistical power to detect adverse outcomes.

This meta-analysis has several limitations. First, all studies reporting long-term outcomes were retrospective non-randomized studies. Thus, a confounding bias could be present. Also, information bias and measurement errors are common in retrospective analyses. Nevertheless, a clinical epidemiological report has demonstrated that a meta-analysis of well-designed nonrandomized studies of surgical procedures is comparable to RCTs [33]. Second, the included studies were conducted in Asia, particularly, in Japan and Korea. Thus, a geographic discrepancy could exist. Third, the postoperative treatment protocols of primary studies were not consistent for the two groups. Therefore, further clinical trials are needed to arrive at a more definitive conclusion. Finally, we could not investigate the effects of neoadjuvant chemotherapy on patients’ survival due to the lack of thorough information concerning preoperative treatments.

## 5. Conclusions

In conclusion, our results show partial omentectomy, compared with total omentectomy in gastric cancer surgery was associated with non-inferior oncological outcomes and comparable safety outcomes. Thus, routine total omentectomy is not recommended as a standard operation step for early and locally advanced gastric cancer.

## Figures and Tables

**Figure 1 cancers-13-04971-f001:**
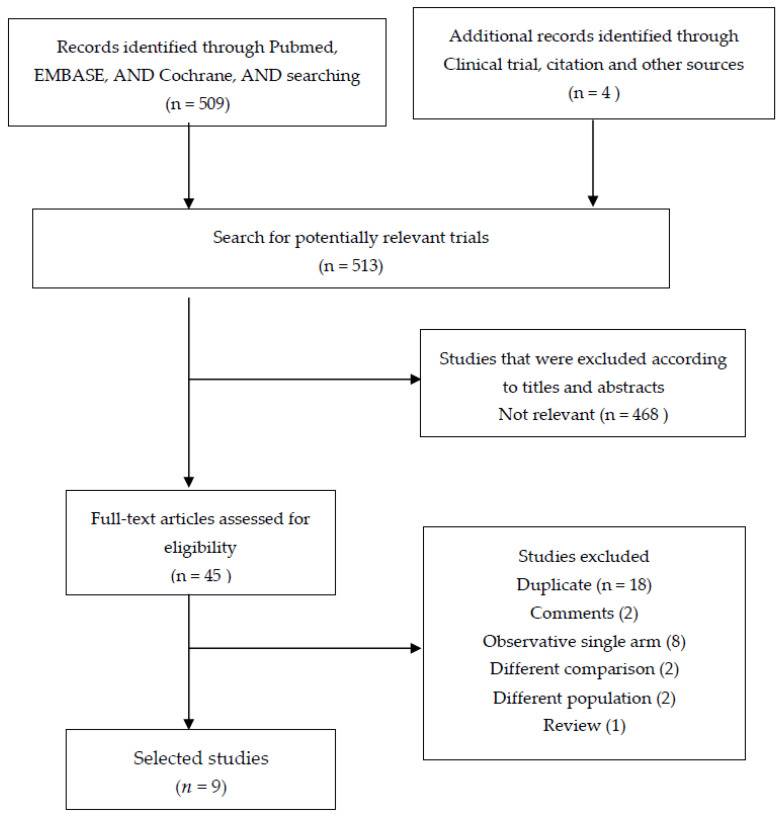
Flow diagram of the articles’ selection process.

**Figure 2 cancers-13-04971-f002:**
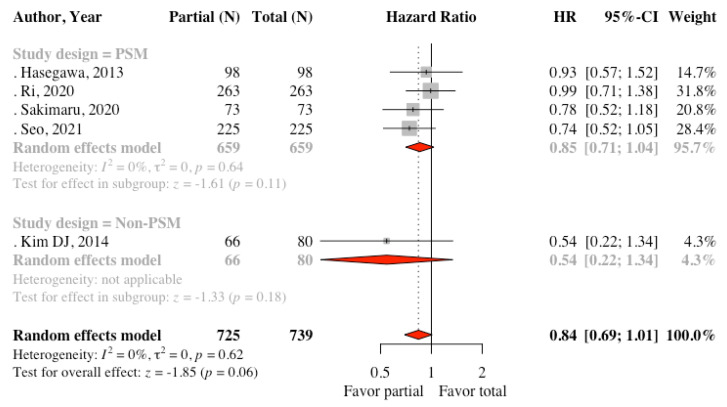
Forest plot comparing disease-free survival between the partial omentectomy group and the total omentectomy group.

**Figure 3 cancers-13-04971-f003:**
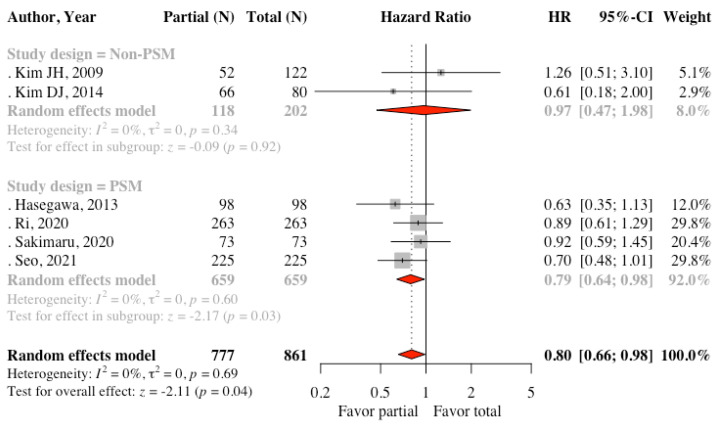
Forest plot comparing overall survival between the partial omentectomy and the total omentectomy group.

**Figure 4 cancers-13-04971-f004:**
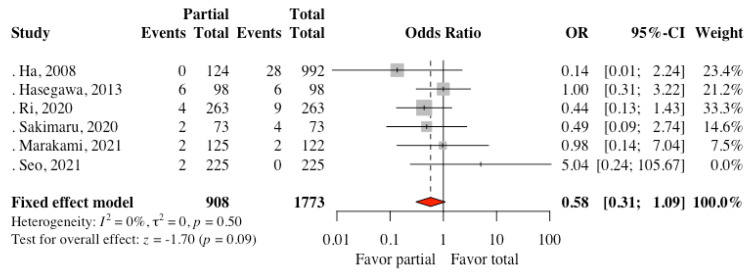
Forest plot comparing adhesion- and ileus-related complications incidence between the partial omentectomy and the total omentectomy groups.

**Table 1 cancers-13-04971-t001:** Characteristics of the selected studies.

Author, Year, Country	Study Design, Inclusion Criteria	Approach	Pathological Staging	Neo/Adjuvant Chemotherapy	Follow up, Months (Median)	Adjustment
Ha, 2008, Korea	-Retrospective -2004–2006 -Early gastric cancer	Open	≥pT3: Partial = 0%; Total = 1.5% LN metastasis: Partial = 7.3%; Total = 7.6%	Not mentioned	20.5 ± 8.6 months (mean + SD) 6.7% loss follow up	None
Kim JH, 2009, Korea	-Retrospective -2000/01–2002/12 -Advanced gastric cancer, pT2	Open	≥ pT3: Partial = 0%; Total = 0% LN metastasis: Partial = 11.8%; Total = 5%	Not mentioned	Not mentioned	None
Kim MC, 2011, Korea	-Retrospective -2005–2006 -Early gastric cancer	Open	≥ pT3: Partial = Not mentioned; Total = Not mention LN metastasis: Partial = 57.7%; Total = 44.3%	Not mentioned	38.1	None
Hasegawa, 2013, Japan	- Retrospective with PSM -2001/01–2009/12 -Advanced gastric cancer (adenocarcinoma, pT2–4, N0–3, M0, R0 resection) -Excluded positive peritoneal lavage	Mixed	≥ pT3: Partial = 65.3%; Total = 69.4% LN metastasis: Partial = 58.2%; Total = 60.2%	Neoadjuvant: Partial = 2/98 Total = 4/98 Adjuvant: Partial = 34/98 Total = 20/98	Partial: 39.6; Total: 61.2	Age, gender, *p*-stage, and extent of lymph node dissection
Kim DJ, 2014, Korea	-Retrospective -2004/07–2011/12 -Serosa negative advanced gastric cancer	Laparoscopic	Stage ≥ IIb: Partial = 42.4%; Total = 41.3% LN metastasis: Partial = 48.5%; Total = 50%	Not mentioned	Not mentioned	None
Ri, 2020, Japan	- Retrospective with PSM -2006 – 2012, Multi-center -Advanced gastric cancer (cT3–4, any N, pR0)	Open	≥ pT3: Partial = 52.5%; Total = 51.0% ≥ pN2: Partial = 44.9%; Total = 43.7%	Adjuvant: Partial = 103/263 Total = 98/263 -chemotherapy with S-1 generally	58.8	28 measurable parameters: Age, sex, TNM stage, tumor location, metastasis in lymph node, etc.
Sakimaru, 2020, Japan	- Retrospective with PSM -2008/03–2017/08 -Advanced gastric cancer (cT3–4) -Excluded M1 including positive peritoneal lavage	Mixed	≥ pT3: Partial = 69.9%; Total = 68.5% LN metastasis: Partial = 57.5%; Total = 69.9%	Adjuvant: Partial = 45/73 Total = 49/73	59	variables from preoperative and perioperative findings that could affect outcomes: cTN stage, lymphadenectomy, etc.
Murakami, 2021, Japan	- Randomized controlled trial (RCT), Phase II -2011/04–2018/10 -Advanced gastric cancer (cT2–4a, N0–2, M0)	Open	>pT3: Partial = 58.4%; Total = 66.4% LN metastasis: Partial = 57%; Total = 64%	Stage II/III disease (except T1N2–3 or T3N0) patients were recommended to receive oral S1	Not mention	None
Seo, 2021, Korea	- Retrospective with PSM -2003/01–2015/12 -Advanced gastric cancer (pT3 or pT4)	Mixed	pT4a: Partial = 50.7%; Total = 55.6% LN metastasis: Partial = 67.6%; Total = 66.7%	Excluded neoadjuvant patient Adjuvant: Partial = 158/225 Total = 149/225	48.2	Patient clinical demographics (age, sex, cT), perioperative outcomes (surgical approach, resection extent, extent of lymph node dissection), and pathologic outcomes (tumor size, T, N)

**Table 2 cancers-13-04971-t002:** Analysis of the methodological quality of the selected studies.

Study	Selection Q1: Representativeness of Exposed Cohort	Selection Q2: Selection of Non-exposed Cohort	Selection Q3: Ascertainment of Exposure	Selection Q4: Outcome of Interest not Present at Start of Study	Comparability	Outcome Q1: Assessment of Outcome	Outcome Q2: Was Follow-Up Long Enough for Outcome to Occur	Outcome Q3: Adequacy of Follow up of Cohort	Overall (Total 9 points)
Ha, 2008	V	V	V	V	V	V			6
Kim JH, 2009	V	V	V	V	V	V			6
Kim MC, 2011	V	V	V	V	V	V		V	7
Hasegawa, 2013	V	V	V	V	VV	V	V	V	9
Kim DJ, 2014	V	V	V	V	V	V		V	7
Ri, 2020	V	V	V	V	VV	V	V	V	9
Sakimaru, 2020	V	V	V	V	VV	V	V	V	9
Seo,2021	V	V	V	V	VV	V	V	V	9
Murakami, 2021	Randomization process: Low risk Deviations from intended interventions: Some concern Missing outcome data: Low risk of bias Measurement of the outcome: Low risk of bias Selection of the reported result: Low risk of bias								Low risk of bias

**Table 3 cancers-13-04971-t003:** Summary measures of primary and secondary outcomes for partial omentectomy versus total omentectomy in gastric cancer patients with subgroup analysis.

Outcome	Pooled Effect (95% CI; *p* Value)	Test for Heterogeneity	Test for Interaction
Overall survival			
Overall (6 studies)	HR: 0.80 (0.66 to 0.98, *p* = 0.04)	*I*^2^ = 0%, *p* = 0.69	NA
Subgroup analysis according to study design			
Analysis without PSM (2 studies)	HR: 0.97 (0.47 to 1.98, *p* = 0.92)	*I*^2^ = 0%, *p* = 0.34	*p* = 0.60
Analysis with PSM (4 studies)	HR: 0.79 (0.64 to 0.98, *p* = 0.03)	*I*^2^ = 0%, *p* = 0.60	
Disease free survival			
Overall (5 studies)	HR: 0.84 (0.69 to 1.01; *p* = 0.06)	*I*^2^ = 0%, *p* = 0.62	NA
Subgroup analysis according to study design			
Study with retrospective (1 studies)	HR: 0.54 (0.22 to 1.34, *p* = 0.18)	NA	*p* = 0.34
Study with PSM (4 studies)	HR: 0.85 (0.71 to 1.04, *p* = 0.11	*I*^2^ = 0%, *p* = 0.64	
Composite outcomes (Clavien-Dindo classification ≥ grade 3)			
Overall (5 studies)	OR: 0.85 (0.60 to 1.21, *p* = 0.37)	*I*^2^ = 50%, *p* = 0.09	
Complication, adhesion and ileus			
Overall (6 studies)	OR: 0.58 (0.31 to 1.09, *p* = 0.09)	*I*^2^ = 0%, *p* = 0.50	
Operative time			
Overall (7 studies)	MD: −16.7 mins (−28.8 to −4.6, *p* < 0.01)	*I*^2^ = 95%, *p* < 0.01	NA
Subgroup analysis according to operation method			
Open method (4 studies)	MD: −2.1 mins (−10.9 to 6.7, *p* = 0.64)	*I*^2^ = 68%, *p* = 0.02	*p* < 0.01
Mixed or MIS method (3 studies)	MD: −32.8 mins (−48.9 to −16.8, *p* < 0.01)	*I*^2^ = 94%, *p* < 0.01	
Estimated blood loss			
Overall (4 studies)	MD: −95.3 mL (−139.8 to −50.8; *p* < 0.01)	*I*^2^ = 96%, *p* < 0.01	
Numbers of Lymph nodes harvested			
Overall (6 studies)	MD: 1.3 (−1.2 to 3.9; *p* = 0.30)	*I*^2^ = 85%, *p* < 0.01	
Length of stay			
Overall (4 studies)	MD: −0.4 (−1.0 to 0.2; *p* = 0.21)	*I*^2^ = 94%, *p* < 0.01	

## Data Availability

The lead author affirms that this manuscript is an honest, accurate, and transparent account of the examined studies; no important aspects of the studies have been omitted, and discrepancies between the studies as planned (and, if relevant, registered) have been explained. Data are contained within the article or Appendix A.

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
