# Peer review of "Partial Versus Total Omentectomy in Patients with Gastric Cancer: A Systemic Review and Meta-Analysis"

_cancers, 2021, doi:10.3390/cancers13194971_

Round 1

Reviewer 1 Report

The paper is well written and detailed; the Authors are confidants and expertics with the topic.

Could the Authors better explain about the papers with cases without notice in pre-operative therapy and the possibility of influencing OS?

Reviewer 2 Report

Introductory part: I would recommend authors to develop multiple roles played by great omentum, along with possibilities to use omentum in several surgical approaches. (cite: Socea B, Constantin V, Carap A, Moculescu C, Costea D, Popa F, Galajda Z. Revascularizarea membrului inferior--între deziderat ÅŸi posibilităţi [Lower limb revascularization--a continuous challenge]. Chirurgia (Bucur). 2011 Sep-Oct;106(5):627-30.). 

Also, I would recommend to mention possible complications in gastric cancer surgery, like fistula formation and to mention biological markers that could announce the possibility of this complication (Radulescu D, Baleanu VD, Padureanu V, et al. Neutrophil/Lymphocyte Ratio as Predictor of Anastomotic Leak after Gastric Cancer Surgery. Diagnostics (Basel). 2020 Oct 9;10(10):799. doi: 10.3390/diagnostics10100799.). 

Discussion part: I would recommend to insert some comments about the extension of partial omentectomy and to define it, compared to total omentectomy. 

Also, I think that it will be interesting to compare and extract some data about laparoscopic versus open omentectomy.
